# Balancing complexity, performance and plausibility to meta learn plasticity rules in recurrent spiking networks

**Basile Confavreux** [ID][1,2*], **Everton J. Agnes** [ID][3], **Friedemann Zenke** [ID][4], **Henning Sprekeler** [ID][5], **Tim P. Vogels** [ID][1]

**1** Institute of Science and Technology Austria, Klosterneuburg, Austria, **2** Gatsby Computational Neuroscience Unit, University College London, London, United Kingdom, **3** Biozentrum, University of Basel, Basel, Switzerland, **4** Friedrich Miescher Institute for Biomedical Research, Basel, Switzerland, **5** Technische Universität Berlin, Berlin, Germany

\* basile.confavreux@gmail.com

**Data availability statement:** The full data and code is publicly available at https://github.com/VogelsLab/SpikES.

## Abstract

Synaptic plasticity is a key player in the brain's life-long learning abilities. However, due to experimental limitations, the mechanistic link between synaptic plasticity rules and the network-level computations they enable remain opaque. Here we use evolutionary strategies (ES) to meta learn local co-active plasticity rules in large recurrent spiking networks with excitatory (E) and inhibitory (I) neurons, using parameterizations of increasing complexity. We discover rules that robustly stabilize network dynamics for all four synapse types acting in isolation (E-to-E, E-to-I, I-to-E and I-to-I). More complex functions such as familiarity detection can also be included in the search constraints. However, our meta learning strategy begins to fail for co-active rules of increasing complexity, as it is challenging to devise loss functions that effectively constrain network dynamics to plausible solutions *a priori*. Moreover, in line with previous work, we can find multiple degenerate solutions with identical network behaviour. As a local optimization strategy, ES provides one solution at a time and makes exploration of this degeneracy cumbersome. Regardless, we can glean the interdependecies of various plasticity parameters by considering the covariance matrix learned alongside the optimal rule with ES. Our work provides a proof of principle for the success of machine-learning-guided discovery of plasticity rules in large spiking networks, and points at the necessity of more elaborate search strategies going forward.

## Author summary

Synapses between neurons in the brain change continuously throughout life. This phenomenon, called synaptic plasticity, is believed to be crucial for the brain to learn from and remember past experiences. However, the exact nature of these synaptic changes remains unclear, partly because they are hard to observe experimentally. Theorists have thus long tried to predict these synaptic changes and how they contribute to learning and

**Funding:** This project has received funding from the HORIZON EUROPE European Research Council (ERC) consolidator grant (SYNAPSEEK, awarded to TV), a Wellcome Trust Sir Henry Dale Research Fellowship (WT100000, awarded to TV), a Wellcome Trust Senior Research Fellowship (214316/Z/18/Z, awarded to TV), and a Sir Henry Wellcome Fellowship (110124/Z/15/Z, awarded to FZ). The funders had no role in study design, data collection and analysis, decision to publish, or preparation of the manuscript.

**Competing interests:** The authors have declared that no competing interests exist.

memory, using abstraction called plasticity rules. Although many plasticity rules have been proposed, there are many different synapse types in the brain and many more possible rules to test. A recent approach has thus been to automate this screening of possible plasticity rules, using modern Machine Learning tools. This idea, called meta learning plasticity rules, has so far only been applied to very simple models of brain synapses. Here, we scale up this idea to more complex and more faithful models. We optimize plasticity rules based on their ability to make model brain circuits solve some basic memory tasks. We find several different yet equally good plasticity rules (degeneracy). However, our method drops in performance when considering more complex rules or tasks.

## Introduction

Synaptic plasticity is thought to be the cornerstone of learning and memory. *In silico*, the evolution of synaptic efficacies is modeled with plasticity rules [1–20] typically derived from *ex vivo* experiments in single synapses [7,21–25]. Even though such rules recapitulate the data gathered at the single neuron level, they often fail to elicit the observed functions or architectures at the network level, in part because of the enormous parameter space that must be trawled to elicit functions such as memory formation in spiking neuronal networks (SNNs) [12,13,16].

Instead of tuning the values of the parameters governing plasticity by hand (hand-tuning), an emerging approach dubbed "meta learning synaptic plasticity" consists in performing numerical optimization on the plasticity rules themselves so that candidate plasticity rules with desired network-level behaviors can be found automatically [26–32]. This approach has been successful in rate networks, both in elucidating the learning rules implemented in brains and proposing alternatives to back-propagation [33–38]. However, in the case of spiking neuronal networks, this meta learning approach has been restricted to two-layer feedforward networks performing simple tasks [31,39]. This dearth is partly owed to the fact that the parameterization of spike-based plasticity rules either involves high-dimensional expressions [6], ill-suited to numerical optimization in spiking networks, or search spaces so simple that they don't contain truly novel rules [31] when used in isolation. Additionally, the non-differentiability of spiking network models and their large compute requirements contribute to the lack of results in meta-learning plasticity rules at the level of large recurrent spiking networks.

Here, we solve some of the above-mentioned difficulties to meta learn biologically plausible plasticity rules in large recurrent spiking networks with excitatory and inhibitory populations in a two-loop meta learning paradigm. In an inner loop, parameterized plasticity rules are embedded in spiking networks performing a given task, while in an outer loop, a Covariance Matrix Adaptation-Evolution Strategy (CMA-ES) [40] adjusts the parameters of the plasticity rules so that the spiking network in the inner loop performs better on the task at hand (Fig 1, see also [31]).

We compare several rule parameterizations; low-dimensional polynomial rules can provide interesting, easily interpretable solutions for multiple co-active rules; plasticity rules parameterized with neural networks (MLPs) allow us to include a richer set of potentially relevant effectors. We show that we can successfully extract suitable rules for a given function, such as stabilizing network dynamics or performing familiarity detection, as long as we focus on a single connection type (e.g., excitatory-excitatory) at a time. When we turn to more complex search spaces, such as co-active rules, or more elaborate tasks, the flexibility of the system makes it very difficult to craft successful loss functions for biologically plausible solutions that

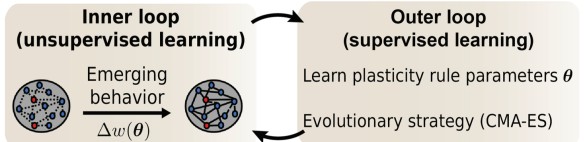

**Fig 1. Meta learning approach to discover plasticity rules with a desired network function.** Plasticity rules that change synaptic weights depending on some biologically plausible synaptic variables—e.g., pre- and postsynaptic spike times—are parameterized with parameters $\theta$. These parameters are optimized using evolutionary strategies to find a plasticity rule that minimizes a loss function quantifying a desired network behavior.

can be learned in finite time. Interestingly, when we find a suitable rule for a given task, we can usually find multiple others, confirming previous results on degenerate solution spaces of plasticity rules [31,41–43].

# Results

The rules governing changes of neuronal connections across time remain an open question in neuroscience, despite decades of efforts. An emerging *in silico* method to propose interesting candidate plasticity rules from network-level constraints—meta learning—has been successful in small feedforward systems [31,39], using low-dimensional plasticity parameterizations [31]. However, it is unclear if this method can scale to larger recurrent spiking networks with more complex plasticity rules. Here, we show that we can meta learn plasticity rules with basic memory functions for a range of plasticity parameterizations. In fact, many different plasticity rules can perform a given computation, i.e., the solution space is degenerate.

## Meta learning procedure

Here, we turned to a previously devised meta learning pipeline (Fig 1): We used CMA-ES [40] to iteratively improve upon an initial plasticity rule, parameterized with plasticity parameters $\theta$. At every meta iteration, CMA-ES generated a collection of plasticity rules which were evaluated in individual spiking networks according to their fitness (loss function). CMA-ES then updated its internal model of parameter interdependencies and its guess for the best rule, i.e., the mean and covariance matrix of a Gaussian distribution in plasticity parameter space $\theta$ (see Methods). To help with the stability of the meta optimization, the initial plasticity rule was chosen such that it elicited no or few weight changes in the network.

## Network stability with a small polynomial search space

We began with a simple search space for plasticity rules encompassing first-order (i.e., containing no square-terms, Methods) spike-timing-dependent plasticity (STDP) rules, which we refer to as the "small polynomial search space." For this search space, similar to previous work [31], the weight from presynaptic neuron $i$ to postsynaptic neuron $j$, $w_{ij}(t)$, evolved as

$$\frac{\mathrm{d}}{\mathrm{d}t}w_{ij}(t) = \alpha S_i(t) + \beta S_j(t) + \gamma S_j(t)x_i(t) + \kappa S_i(t)y_j(t) \tag{1}$$

where $S_i(t)$ is the spike train of neuron $i$, $x_i(t)$ is a low pass filters of the spike train of the pre-synaptic neuron $i$ with time constant $\tau_{\mathrm{pre}}$, and $y_j(t)$ is a low pass filters of the spike train of the post-synaptic neuron $j$ with time constant $\tau_{\mathrm{post}}$. The spike train is defined as

$S_i(t) = \sum_k \delta(t - t_k^i)$, where $t_k^i$ is the time of the k-*th* spike of neuron $i$ and $\delta$ is the Dirac delta. In total, this search space comprised six tunable plasticity parameters: $\theta = [\alpha, \beta, \gamma, \kappa, \tau_{pre}, \tau_{post}]$.

Previous work uncovered inhibitory to excitatory (I-to-E) rules that would enforce network stability in a feedforward setting [31]. To discover such rules in recurrent networks, we meta-learned I-to-E plasticity rules that enforced a target population firing rate of 10 Hz, quantified with a loss function on the network activity (see Methods, Fig 2A, "stability task").

The ES converged to a rule with low loss values such that network activity remained stable at 10 Hz for twice the longest possible training duration (2 min, Fig 2C). Visualizing the meta-learned rule with a classic pre-post protocol (pairs of pre-/postsynaptic spikes with various delays, Fig 2B, left) revealed that it did not correspond to any known rules [44]. The

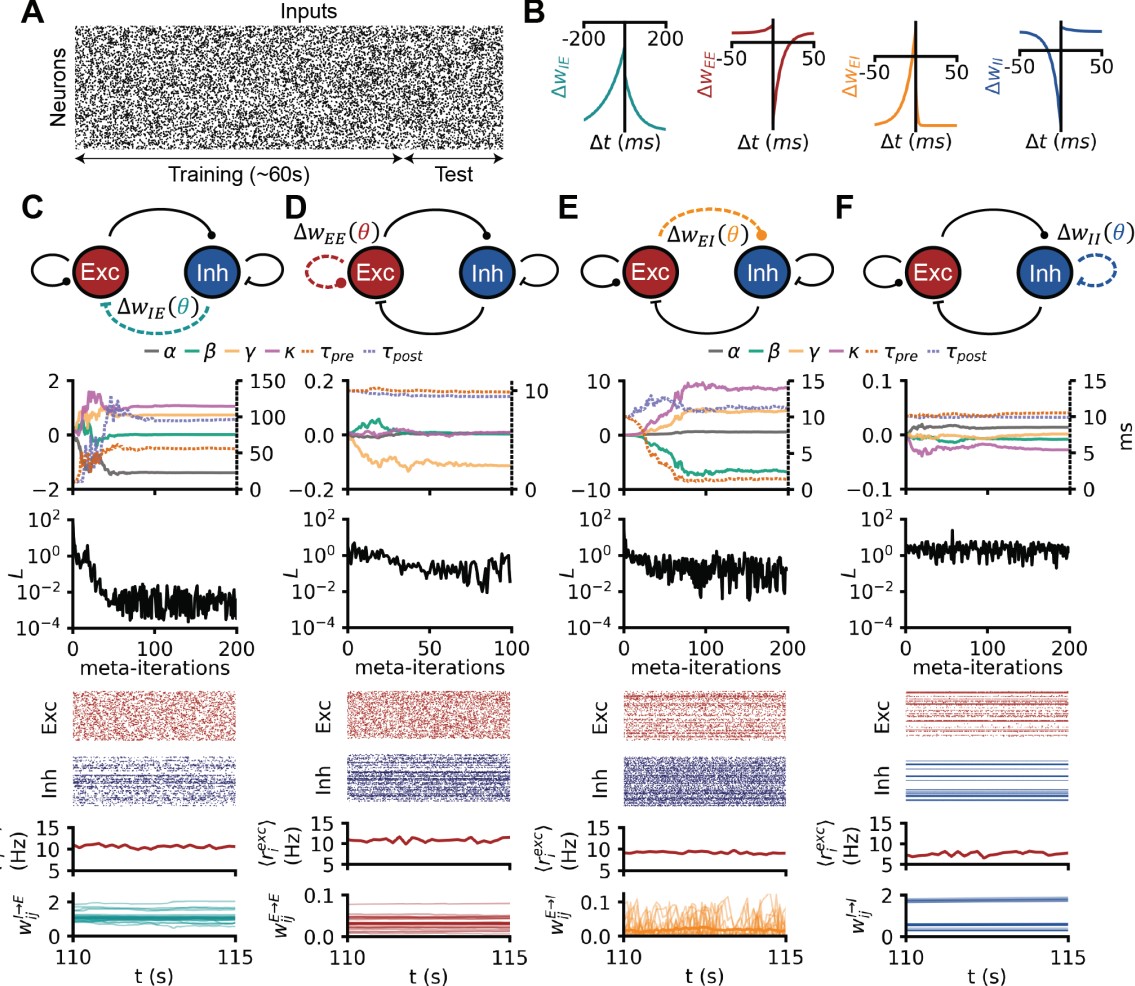

**Fig 2. Network stabilization with simple polynomial rules in isolation.** (A) Raster plot of inputs received by a recurrent spiking network undergoing the stability task. (B) Pre-post protocols of the four (separately) meta learned plasticity rules in C, D, E and F. (C) A spiking network received Poisson input at a random rate, the E-to-E synapses are plastic with a rule from the small polynomial search space. From top to bottom: (i) evolution of the 6 plasticity parameters during meta learning with CMA-ES. (ii) Evolution of the loss during meta-optimization. (iii) Raster plot of the 200 random excitatory neurons of a network evolving with the final meta learned I-to-E rule. (iv) same as (iii) for the inhibitory neurons. (v) evolution of the population firing rate of excitation (vi) evolution of E-to-E weights (thicker line: mean). D: Same as C, but for E-to-I plasticity. E: Same as C, but for I-to-E plasticity. F: Same as C, but for I-to-I plasticity.

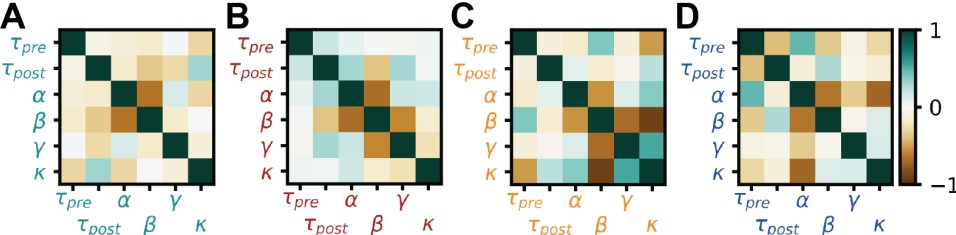

**Fig 3. Interpretation of meta learned rules for stability.** (A) Covariance matrix at meta-iteration 15 of the optimization in Fig 2C (See also Supplementary Materials and S2 Fig). (B) Same as A for the optimization shown in Fig 2D. (C) Same as A for the optimization shown in Fig 2E. (D) Same as A for the optimization shown in Fig 2F.

rule resembled the symmetric inhibitory rules found in theory and experiments [10,24], but shifted downward such that all pairs of pre-post spikes elicited depression. Note that since in our simulations the number of pre- and post- synaptic spikes (the pre- and postsynaptic firing rates) do not have to be the same, this graph does not mean that the meta learned rule has no potentiation regions (Fig 2B). In the network simulation, the I-to-E weights do settle to intermediate values as a result of both potentiation and depression (Fig 2C, bottom).

Next, we used the same stability constraint to discover plasticity rules in other synapse types (E-to-E, E-to-I, or I-to-I, individually, Fig 2D–2F). In two scenarios—E-to-E and E-to-I—the optimization was able to find rules that established the target firing rate (Fig 2C and 2D), albeit not as effectively as with I-to-E plasticity, as evidenced by relatively high losses at the end of training.

Since all successful rules differed from previously observed rules (Fig 2B), we wanted to understand the inter-dependencies of the learned plasticity parameters. We thus plotted the covariance matrix between plasticity parameters as the meta learned rules emerged during the optimization. The covariance matrix is updated alongside the optimal rule in CMA-ES and contains information about the loss landscape, albeit heuristically. Our analysis revealed that the main structure in the rules was strongly anti-correlated non-Hebbian plasticity parameters, i.e., when $\alpha$ is increased $\beta$ is likely decreased and vice versa (Fig 3), consistent with mean-field theory (see Mean-field analysis). For I-to-I plasticity, no plasticity rule could be found that improved meaningfully upon the initial (no plasticity) rule (Fig 3D), although we note that the absence of proof is not proof of the absence of a solution.

## Familiarity detection with a small polynomial search space

Having established that rules from the small polynomial search space could stabilize recurrent spiking networks, we turned to a more complex, memory-related network function: familiarity detection. This ubiquitous form of memory [45,46] has already been the target of meta learning in rate networks [33], and has been shown to emerge in recurrent spiking networks with finely orchestrated, hand-tuned co-active synaptic plasticity rules [12,13,16].

To meta learn plasticity rules that would produce familiarity detection, we designed a protocol in which we stimulated a recurrent spiking network with the same stimulus multiple times, which we define as the "familiar" stimulus. After extensive stimulation with this familiar stimulus intended to induce strong changes in synapses, we compared network responses with a non-overlapping second stimulus that we define as "novel". Successfully learning the familiar stimulus meant responding to it with a higher firing-rate compared to the novel stimulus after learning. To accomplish such a learning, we designed a loss function that

constrained plasticity rules such that the network would produce high firing rates to familiar, and low firing rates to novel stimuli (see Methods and S1 Fig).

We started by optimizing only I-to-E plasticity while all other rules were inactive. We refer to such scenarios as single-active rules. The I-to-E plasticity rule belonged to the small polynomial search space. Our ES algorithm converged to a rule that achieved low loss values (Fig 4C) and produced networks that responded more strongly to familiar than to novel

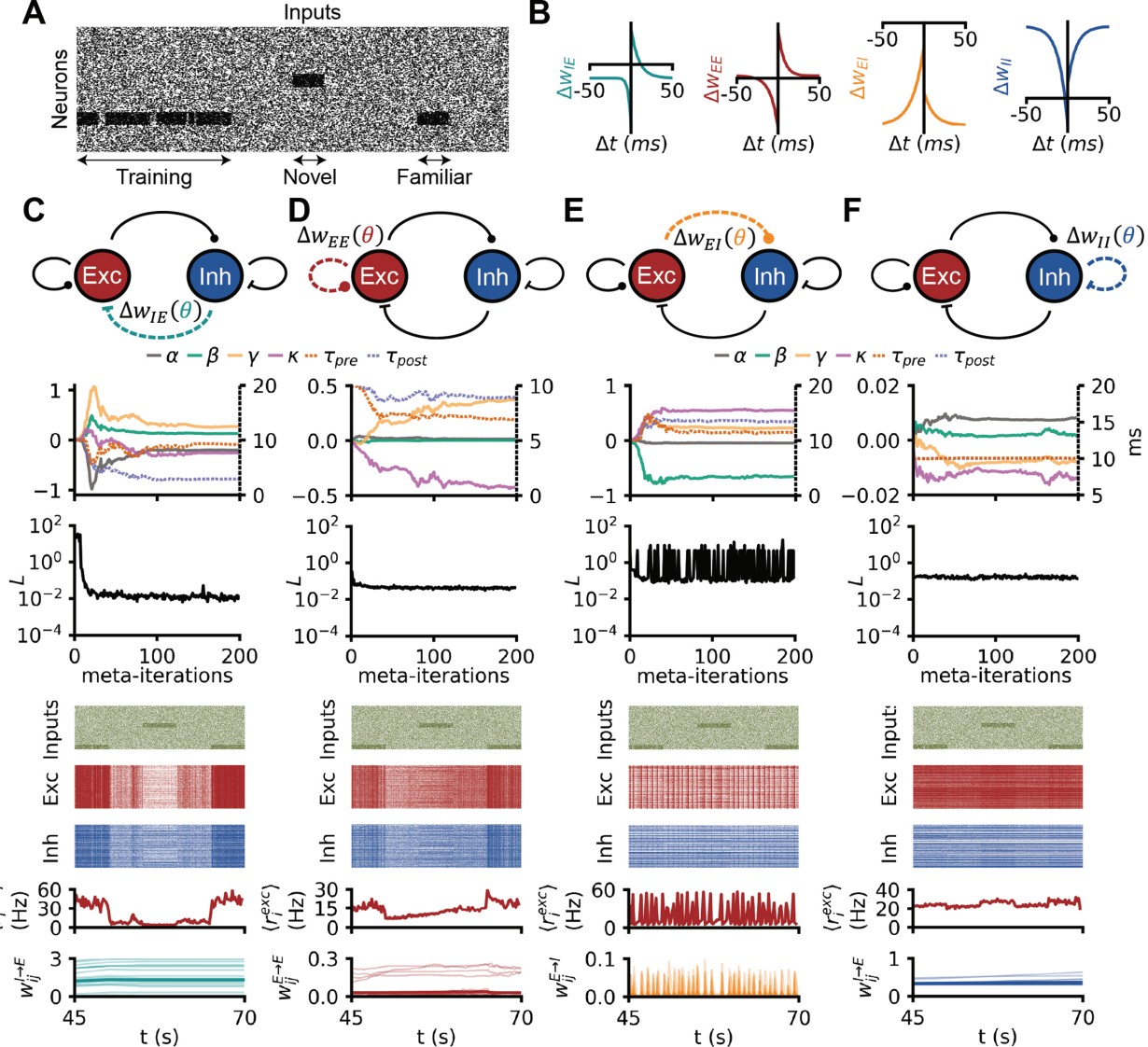

**Fig 4. Familiarity detection with simple plasticity rules.** (A) Raster plot of inputs received by a recurrent spiking network undergoing the familiarity task: the network is trained on a familiar stimulus, then after a break the network is shown a novel stimulus and the familiar stimulus. (B) Pre-post protocols of the four (separately) meta learned plasticity rules in C, D E and F. (C) A spiking network undergoing the familiarity task: the E-to-E synapses are plastic with a rule from the small polynomial search space. From top to bottom: (i) evolution of the 6 plasticity parameters during meta learning with CMA-ES. (ii) Evolution of the loss during meta optimization. (iii) Raster plot of the 200 random excitatory neurons of a network evolving with the final meta learned I-to-E rule. (iv) same as (iii) for the inhibitory neurons. (v) evolution of the population firing rate of excitation (vi) evolution of E-to-E weights (thicker line: mean). D: Same as C, but for E-to-I plasticity. E: Same as C, but for I-to-E plasticity. F: Same as C, but for I-to-I plasticity.

stimuli. As a control, a network undergoing the same task without any plasticity was unable to exhibit asymmetric responses for novel versus familiar stimuli, confirming that the learned plasticity rule was responsible for this acquired behavior (S5 Fig). When we probed the plasticity rule with classical pre/post protocols, we found that the rule did not closely resemble any of the experimentally reported temporal relationships [44]. Notably, familiarity detection was achieved here with a single active I-to-E plasticity rule, contrary to previous work in which memory-related functions were always achieved *in tandem* with E-to-E plasticity [12, 13,16,47].

Next, we focused on E-to-E plasticity in isolation, using the same loss function. The ES found an E-to-E plasticity rule that was able to solve the familiarity task (Fig 4D), and its pre-post protocol resembled previously reported classical asymmetric STDP rules [21,22]. We also considered the other two synapse types (E-to-I, or I-to-I, individually active), but ES could not find satisfying solutions for either (Fig 4E and 4F). Note again, that this result does not prove that no solutions exist within the small polynomial search space for E-to-I or I-to-I rules.

The covariance matrices corresponding to the optimizations for the familiarity task were somewhat similar to the ones obtained on the stability task ($\alpha$-$\beta$ anti-correlations for Figs 3A, 3B and 5A, 5B; $\beta$-$\gamma$ and $\beta$-$\kappa$ anti-correlations as well as $\gamma$-$\kappa$ correlation for Figs 3C and 5C; $\alpha$-$\beta$, $\alpha$-$\gamma$ and $\alpha$-$\kappa$ anti-correlations for Figs 3D and 5D), suggesting similar structure in the relationships of the learned rules for both cases (strong anti-correlation between non-Hebbian parameters, Fig 5).

## Familiarity detection with co-active simple polynomial rules

Inspired by previous work proposing co-active E-to-E and I-to-E rules for memory formation in spiking networks [12,13,16], we set out to meta learn jointly the E-to-E and the I-to-E plasticity rules for the familiarity detection task mentioned above (Fig 6). Since either rule (I-to-E or E-to-E) was shown to be able to solve this task individually, ES should succeed in finding at least one solution. As expected, ES converged on a solution that satisfied all constraints and displayed the hallmarks of cortical network dynamics. The learned E-to-E rule was similar to the above-described rule acting in isolation (Fig 4C), displaying a very similar shape as experimentally observed E-to-E rules [3,21,22]. The newly learned I-to-E rule, on the other hand, differed from previous experimental results [7,10,23,48] and also from the previous optimization above, showing an inverse, asymmetric "Bavarian Hat" with a tuft of potentiation. The covariance matrix revealed anti-correlation between non-Hebbian terms of each rule, and

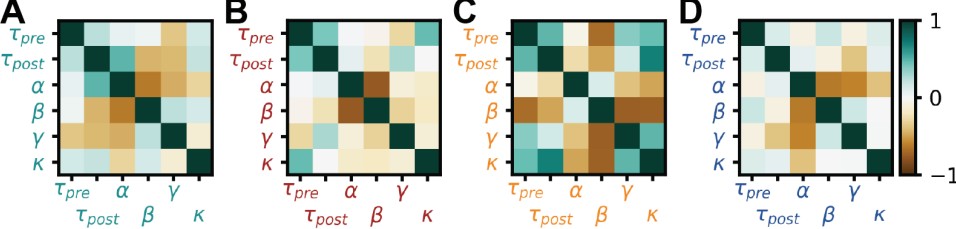

**Fig 5. Interpretation of meta learned rules for familiarity detection.** (A) Covariance matrix at meta-iteration 15 of the optimization in Fig 4C (B) Same as A for the optimization shown in Fig 4D. (C) Same as A for the optimization shown in Fig 4E. (D) Same as A for the optimization shown in Fig 4F.

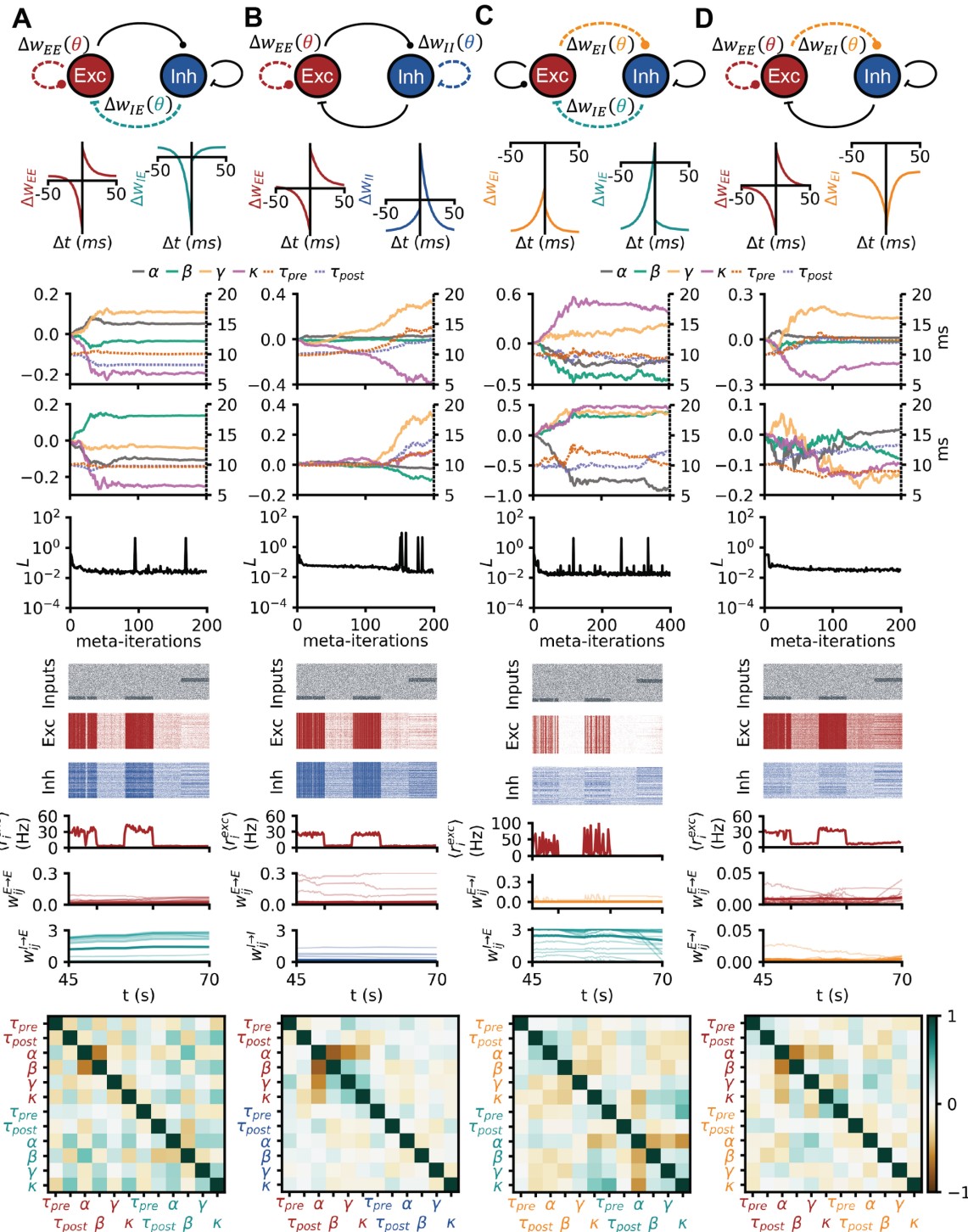

**Fig 6. Familiarity detection with simple co-active rules.** (A) Same network and familiarity task as in Fig 3, but both the E-to-E and I-to-I weights are plastic with rules from the simple polynomial search space. From top to bottom: network diagram, pre-post protocols of the 2 optimized co-active rules, evolution of the 6 parameters for both rules across the optimization, evolution of the loss across meta-training, covariance matrix at meta-iteration 20. (B) Same as A for a network with tunable E-to-I and I-to-I rule. (C) Same as A for a network with tunable E-to-E and E-to-I rule. (D) Same as A for a network with tunable E-to-E and I-to-E rule.

some interactions between parameters of both rules, namely a inverse relationship between non-Hebbian parameters (Fig 6, bottom row).

We also tried other combinations of co-active rules on the same task (E-to-E and I-to-I, E-to-I and I-to-E, as well as E-to-E and E-to-I). In all cases, ES converged to solutions that elicited higher responses to familiar than to novel stimuli, but the network dynamics were biologically implausible (Fig 6), suggesting that E-to-E and I-to-E were the most useful synapse-type for the considered function. Alternatively, it could be that the plasticity rules we used were not flexible or broad enough to express biologically plausible solutions.

## More complex plasticity rules

To capture more complex plasticity mechanisms we constructed two higher dimensional and more expressive plasticity rule parameterizations, i.e., (1) a polynomial with additional dependencies and (2) a neural network parameterization ("MLP", a feedforward network that determines the synaptic changes of the recurrent spiking network, see Methods). We benchmarked these two new parameterizations on the same stability task as for the small search space (Fig 2).

First, we expanded the small polynomial search space, adding synaptic variables that contributed to weight updates, such as additional synaptic traces (triplets rules [8], bursts [17], voltage dependence [9], codependent plasticity [20] and weight dependence [5]),

$$
\begin{aligned}
\frac{\mathrm{d}}{\mathrm{d}t} w_{ij}(t) = S_i(t) f_{pre} \left( x_{i,\tau_{\text{short}}}, x_{i,\tau_{\text{long}}}, x_{i,\tau_{\text{long}}}, x_{j,\tau_{\text{short}}}, w_{ij}, \langle V_j \rangle, C_{j,E}, C_{j,I} \right) \\
+ S_j(t) f_{post} \left( x_{i,\tau_{\text{short}}}, x_{i,\tau_{\text{long}}}, x_{i,\tau_{\text{long}}}, x_{j,\tau_{\text{short}}}, w_{ij}, \langle V_j \rangle, C_{j,E}, C_{j,I} \right),
\end{aligned}
\tag{2}
$$

where $S_i(t)$ and $S_j(t)$ are the spike times of pre- and postsynaptic neurons, respectively, $f_{pre}$ and $f_{post}$ are polynomial functions with the following synaptic variables: $x_{i,\tau_{\text{long}}}(t)$ and $x_{i,\tau_{\text{short}}}(t)$ are low pass filters of the spike train of the pre-synaptic neuron $i$ with time constants $\tau_{\text{short}} = 10$ ms and $\tau_{\text{long}} = 100$ ms (and similarly for post-synaptic neuron $j$); $C_{j,E}(t)$ and $C_{j,I}(t)$ are co-dependent terms representing the activity of neighboring synapses, i.e., low-pass filtered with fixed time constants $\tau_{C^E} = 10$ ms and $\tau_{C^I} = 100$ ms, as in previous work [20]; and $\langle V_j(t) \rangle$ is the low-pass filtered membrane potential, with a time-constant $\tau_{\langle V \rangle} = 100$ ms [9]. We assumed separability of the synaptic variables, i.e., that the synaptic variables contributed independently to weight updates, which allowed us to incorporate additional dependencies to the weight updates without bloating the total number of plasticity parameters. Meta-learning I-to-E rules on the stability task in the larger polynomial search space resulted in solutions that achieved low losses (Fig 7A). However, when simulating the learned rule for longer than during training, we observed that the rule did not generalize as well as the rules from Fig 2A, with the excitatory activity showing large oscillations around the desired target of 10 Hz (Fig 7A). Additionally, some I-to-E weights reached the maximum weight (10, Fig 7A). The covariance matrix revealed a much sparser structure. The shape of the rule was ambiguous under the pre-post protocol, as it did not constrain the values of the additional synaptic variables.

Motivated by previous work proposing co-active rules that support memory formation and recall in spiking networks [12,13], we considered the same familiarity task and loss function as above, with E-to-E and I-to-E synapses plastic parameterized with the bigger polynomial search space (Fig 7G). Once again, we could meta learn rules that solved the task (Fig 7G). We verified that the learned co-active rules were able to elicit different population responses to the familiar and novel stimuli (Fig 7F). However, other aspects of network activity that were

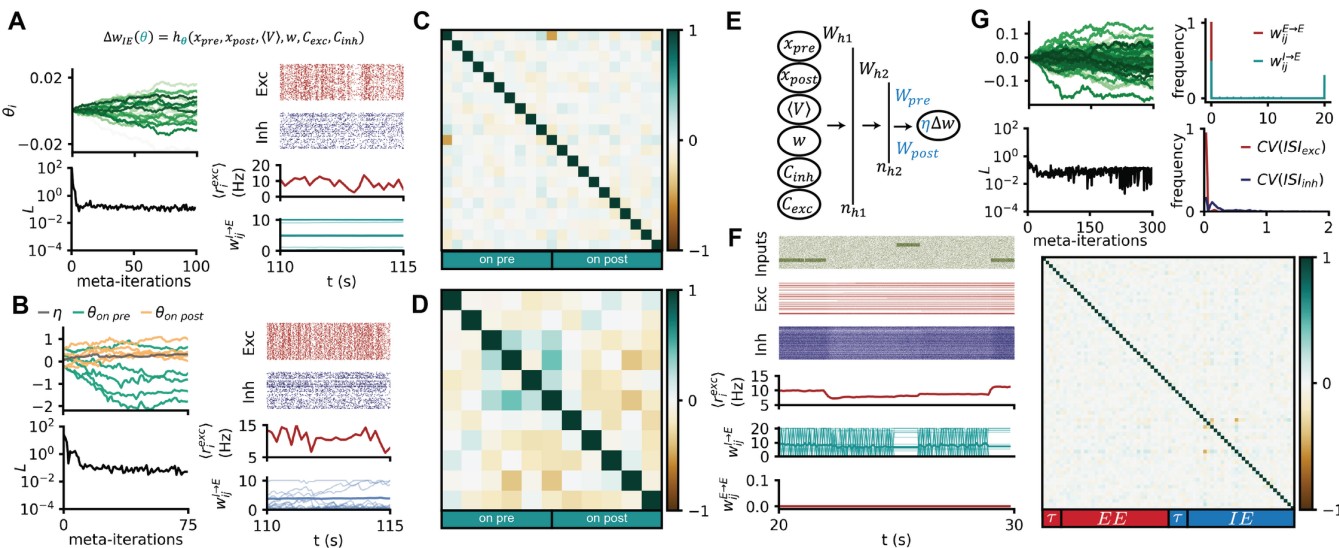

**Fig 7. Meta learning complex plasticity rules with ES.** (A) CMA-ES on I-to-E plasticity within the big polynomial search space for the stability task. Left: Evolution of the plasticity parameters and loss values along the optimization. Right: example network activity elicited by the meta learned I-to-E rule. (B) Same as A for I-to-E rules from the MLP search space. (C) Covariance-matrix of the plasticity parameters for the optimization shown in A. (D) Covariance-matrix of the plasticity parameters for the optimization shown in B. (E) Schematics of the MLP search-space: weight updates in the spiking network are computing by running forward an MLP with synaptic variables as inputs. (F,G) CMA-ES on E-to-E and I-to-E plasticity within the big polynomial search space for the familiarity detection task. G, left: Evolution of the plasticity parameters and loss values along the optimization. G, right and F: example network activity elicited by the meta learned I-to-E rule. F, right: Covariance-matrix of the plasticity parameters for the optimization shown in G.

not constrained by the loss function were unrealistic. For example, most neurons in the network were either silent or fired at unrealistic rates with in highly regular patterns (Fig 7F and 7G). The E-to-E connections mostly converged to zero weights (Fig 7F and 7G). The I-to-E connections underwent rapid switching between 0 and the maximum allowed weight at the millisecond scale, resulting in a bimodal distribution (Fig 7F and 7G).

Finally, we considered a neural-network-based search space for plasticity rules, in which the same synaptic variables as in the big polynomial were combined using a (generalized) multilayer perceptron (MLP, Fig 7B) [49]. Plastic synapses from this search-space underwent spike-triggered updates such that:

$$\frac{\mathrm{d}}{\mathrm{d}t}w_{ij}(t) = S_i(t)\mathrm{MLP}_{\mathrm{pre}}\left(x_{i,\tau_{\mathrm{long}}}, x_{j,\tau_{\mathrm{short}}}, w_{ij}, \langle V_j\rangle, C_{j,E}, C_{j,I}\right) + \\ S_j(t)\mathrm{MLP}_{\mathrm{post}}\left(x_{i,\tau_{\mathrm{short}}}, x_{j,\tau_{\mathrm{long}}}, w_{ij}, \langle V_j\rangle, C_{j,E}, C_{j,I}\right),$$

(3)

using similar notations as for the big polynomial search-space (see More complex plasticity rules).

Similarly to the case with a single connection being plasticity, the MLP used to model synaptic changes taking place in the recurrent spiking network was a feedforward network with two hidden layers (50 and 4 hidden units, see Methods), in which only the final layer weights and bias were tunable, keeping all other layers fixed [49]. This design choice effectively decoupled the number of synaptic variables involved in the rule and the number of plasticity parameters to optimize, thus allowing for potentially highly non-linear dependencies on the synaptic variables while keeping the dimensionality of the search space as low as desired for the evolutionary strategy. This search space comprised a total of 11 parameters:

5 parameters for updates triggered by presynaptic spikes, 5 for postsynaptic updates, and a common learning rate (Fig 7E).

The evolutionary strategy was able to find plasticity rules that established the target firing rate in the MLP search space (Fig 7B). Similar to the big polynomial case, however, the learned rule was not as robust when tested on longer-than-training durations. In addition, the rule elicited biologically implausible network behaviours, for example, weights reaching the maximum allowed value and synchronous spiking patterns (Fig 7B).

Overall, all three parameterizations—small polynomial, big polynomial and MLP—led to rules that solved the task as it was quantified by the loss function. However, the meta learned rules from larger plasticity search spaces did not generalize as well as the simpler rules, and the resulting plastic networks exhibited implausible behaviors, such as synchronous regular firing patterns and bimodal distributions. In our hands, designing a loss function that constrained task performance alone was not sufficient to ensure that biologically relevant plasticity rules emerged.

### Interpreting learned rules and degeneracy

Concerned by the impact of potential degeneracy on the rules proposed in this study, we set to test how reliable our rule predictions were on the familiarity task. Running two (intrinsically stochastic) evolutionary searches from the same starting point on the familiarity task with an I-to-E small polynomial rule converged to two plasticity rules with dissimilar pre-post protocols (Fig 8A). This shows, in agreement with previous work in rate networks [41], that at least two and probably many plasticity rules from the same search space can solve this task. This conclusion is not unique to I-to-E plasticity (Fig 8B). However, even though the plasticity rules differed across optimizations, the relationship between plasticity parameters appeared to be conserved. For example, we observed strong anti-correlations between non-Hebbian parameters in all simulations, as shown by the covariance matrices (Fig 8).

## Discussion

In this study, we scaled up the automatic tuning of plasticity rules for homeostatic and memory-related tasks from single spiking neurons to large recurrent spiking networks. We used an evolutionary strategy to adjust flexibly parameterized plasticity rules in several search spaces and showed the potential and limitations of this gradient-free meta learning approach for *in silico* plasticity rule discovery.

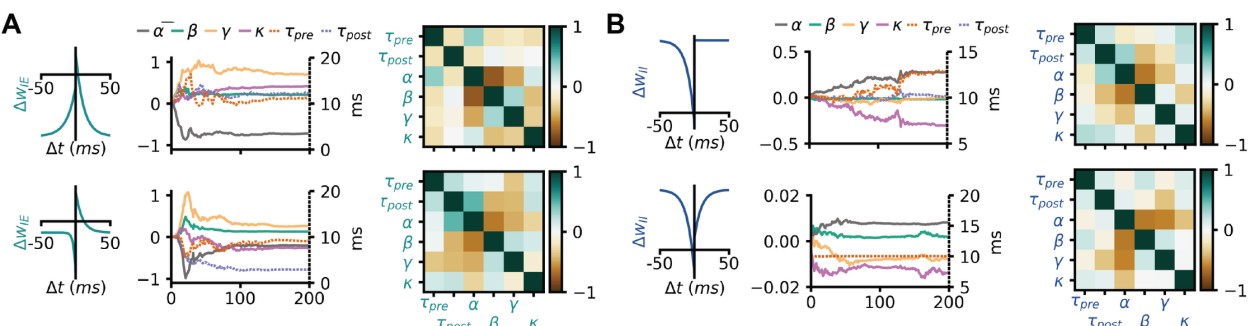

**Fig 8. Degeneracy and solution manifolds.** (A) Bottom: optimization for an I-to-E small polynomial rule on the familiarity task shown in Fig 4C. Top: pre-post protocol, parameter evolution and covariance matrix of another optimization for an I-to-E small polynomial rule on the familiarity task. (B) Same as A, for an I-to-I small polynomial rule on the familiarity task.

As expected from previous work [10,11,15,31,49], we could find isolated I-to-E plasticity rules that enforced firing rate homeostasis (Fig 2). Similar homeostatic network effects could be achieved with isolated E-to-E and E-to-I rules (Fig 2). In our hands, no I-to-I plasticity rule could be found to serve rate homeostasis (Fig 2). We assumed that homeostasis by way of I-to-I synapses is more difficult to achieve because they can only affect the rates of excitatory neurons indirectly (see Mean-field analysis).

To get a better understanding of the meta learned plasticity rules, we made use of the covariance matrix as it emerged during meta learning with CMA-ES [40]. We interpreted deviations from zero in this matrix as acquired biases in the sampling of new plasticity rule candidates, i.e., the matrix unveiled likely interdependencies between parameters that led to successful plasticity action; strong anti-correlations between non-Hebbian parameters was a hallmark of all successful rules (Fig 3). We interpreted these inverse relationships to imply that even for Hebbian-seeming rules a substantial part of the weight changes were effected by pre-only and post-only–i.e., non-Hebbian–terms.

Next we searched for rules that could perform a computational task. We chose familiarity detection as a fundamental component of any memory function beyond mere activity homeostasis. Isolated E-to-E rules that were sufficient for this function could be found (Fig 4), in contrast to previous work that requires several co-active, finely orchestrated rules [12,13]. However, we did not check the long-term stability of the representations achieved by our rules, nor the emergence of attractor dynamics, which may require additional complexity, or additional types of rules. Moreover, isolated I-to-E plasticity rules could also be found to solve this memory task, hinting at a greater role for I-to-E plasticity, with a wide range of potential functions other than network stabilization [48,50]. Isolated E-to-I and I-to-I rules could not be be found to store patterns in spiking networks, justifying *post hoc* the relative dearth of previous modeling studies on the function of these synapse types. Of course, the absence of proof does not prove the absence of solutions that could perform familiarity detection with isolated E-to-I or I-to-I plasticity. Remarkably, the parameter interdependencies for the familiarity task as revealed by the meta learned covariance matrix were similar to the ones for the stability task (Figs 3 and 5), suggesting that the same plasticity mechanisms could enforce homeostasis *and* support basic memory functions [49].

When we broadened our search spaces to successfully meta learn multiple co-active plasticity rules with similar basic memory functions (Fig 6), our joy in finding these sets of rules was somewhat tempered by the fact that isolated individual rules could solve the task at hand already, but our results provided a proof of principle for the possibility of discovering ensembles of co-active rules [49].

We also broadened the complexity of how we parametrized individual rules, going from simple polynomials [31] to expressions with more synaptic variables using either larger polynomials, or MLPs (Fig 7). Such an expansion of the search space did not scale well with regards to the compute requirements of the outer loop. We thus proposed a partially tunable MLP as a means to non-linearly mix multiple synaptic variables without bloating the parameter number, in line with previous work [49]. The added complexity in the rule space resulted in decreased robustness and generality of the meta learned rules (Fig 7), suggesting that the loss function was not constrained enough for these more flexible rules (Fig 7). Therefore, though the plasticity rules were automatically tuned by way of meta learning, the loss function now required extensive hand-tuning to effectively force network activity and weight dynamics into plausible regimes. To summarize, the pitfalls of the ES were, in our hands, (1) the known performance drop of CMA-ES in higher dimensions, (2) an exponential increase in compute time with increased dimensionality (more and more rules need to be tested per meta-iteration), and (3) the need to restart optimization from scratch upon defining a new

loss function. These three factors rendered hyperparameter optimization and debugging increasingly impractical. For meta learning to be able to scale to large plasticity search spaces and complex network models, we developed an approach elsewhere [49] that alleviates the problem of having to define *a priori* a loss function that controls both task performance and biological plausibility. Here, with ES, we can only guess the full loss function beforehand and subsequently identify flaws in these constraints by inspecting the "optimized" networks *post hoc*; refinement of the loss function then required us to restart the optimization from scratch, at great computational cost (Fig 7).

Moreover, parameters not directly related to synaptic plasticity also have an impact on the meta learned rules. For instance, the initial connectivity can render some plasticity rules stable or unstable [51]. Here, we evaluated each plasticity rule in several networks, with random initial mean weights, connectivity distributions, and input rates. The goal of this averaging was to ensure that we discovered rules that achieved low losses on a range of inputs and initial weight connectivities. However, such averaging could not explore exhaustively more fine-grained initial connectivity motifs [51]. On the other hand, systematically testing connectivity-rule pairs would bloat the parameter space substantially, and render meta learning computationally unfeasible.

Previous work highlights degeneracy of mechanisms in neuroscience and more recently in synaptic plasticity [41–43,49,52]. We confirm these findings, although our ES approach is ill-suited to explore more thoroughly degeneracy due to its local search nature. Notably, we show that degeneracy emerges already in the simplest case (single small polynomial rule on the stability task, Fig 8), which may hint at degeneracy as a ubiquitous phenomenon on plasticity.

Understanding the meta learned rules is challenging, especially in high-dimensional search spaces. In the simpler case of stabilization with the small polynomial search space, we could rely on a predicted subspace of solutions using mean-field theory (see Mean-field analysis, S2 FigC). In the bigger search spaces presented in this work, understanding the resulting learning rules and their relationships to other rules is important to be able to formulate experimental predictions. However, despite the use of an L1 regularization in all optimizations in this study, most meta learned rules still comprised many non-zero parameters (Fig 7) that made a direct comparison to experimental results challenging.

We proposed that the covariance matrix learned with CMA-ES alongside the best rule could help reveal structure in the meta learned parameters. The covariance matrix from an optimization on the stability task in the small polynomial search space is in agreement with insights from mean-field theory (see Mean-field analysis), in that the task is mainly solved by the non-Hebbian terms (Fig 3). The covariance matrix from optimizations of more complex parameterizations is much sparser than the meta learned solution (Fig 6 and 7) and suggests that a few terms are of special importance for this solution.

Overall, we believe that meta learning approaches for synaptic plasticity face a compromise: simple search spaces are easier to optimize, yet their simplicity makes them already amenable to theoretical analysis and often means no truly novel rules can be discovered. Here, meta learning with genetic search algorithms was successful albeit in very limited realms. On the other hand, it served as a first step and spawned a number of new approaches to automatically scan the uncharted depth of plasticity in the future.

## Materials and methods

### Neuron and network model

We considered recurrent networks of excitatory and inhibitory, conductance-based leaky-integrate-and-fire neurons. Two types of networks were implemented, emulating either the networks used by Vogels, Sprekeler et al. [10] or by Zenke et al. [13].

**Networks following Vogels, Sprekeler et al. [10].** This network comprised 8000 excitatory and 2000 inhibitory neurons. The membrane potential dynamics of neuron $j$ (excitatory or inhibitory) were given by

$$\tau_m \frac{\mathrm{d}}{\mathrm{d}t} V_j(t) = -\left(V_j(t) - V_{\text{rest}}\right) - g_j^{\text{AMPA}}(t)\left(V_j(t) - E_{\text{AMPA}}\right)$$
$$- g_j^{\text{GABA}}(t)\left(V_j(t) - E_{\text{GABA}}\right), \tag{4}$$

with $\tau_m$ = 20 ms, $V_{\text{rest}}$ = –60 mV, $E_{\text{AMPA}}$ = 0 mV and $E_{\text{GABA}}$ = –80 mV. A postsynaptic spike was emitted whenever the membrane potential $V_j(t)$ crossed a threshold $V^{\text{th}}$ = –50 mV, with an instantaneous reset to $V_{\text{rest}}$ for the duration of the refractory period, $\tau_{\text{ref}}$ = 5 ms.

The excitatory and inhibitory conductances, $g^{\text{AMPA}}$ and $g^{\text{GABA}}$ evolved such that:

$$\frac{\mathrm{d}}{\mathrm{d}t} g_j^{\text{AMPA}}(t) = -\frac{g_j^{\text{AMPA}}(t)}{\tau_{\text{AMPA}}} + \sum_{i \in \text{Exc}} w_{ij}(t) S_i(t)$$
$$\frac{\mathrm{d}}{\mathrm{d}t} g_j^{\text{GABA}}(t) = -\frac{g_j^{\text{GABA}}(t)}{\tau_{\text{GABA}}} + \sum_{i \in \text{Inh}} w_{ij}(t) S_i(t) \tag{5}$$

with $\tau_{\text{AMPA}}$ = 5 ms, $\tau_{\text{GABA}}$ = 10 ms, $w_{ij}(t)$ the connection strength between neurons $i$ and $j$ (unitless), $S_i(t) = \sum \delta(t - t_i^*)$ the spike train of presynaptic neuron $i$, where $t_i^*$ denotes the spike times of neuron $i$, and $\delta$ the Dirac delta. Unless mentioned otherwise, all neurons received input from 5000 Poisson neurons, with 5% random connectivity and constant rate $r_{\text{ext}}$ = 7 Hz. The recurrent connectivity was instantiated with random sparse connectivity (2%).

This network was used for Figs 2 and 3. All other simulations used the network model described below.

**Networks following Zenke et al. [13]** This network comprised 4096 excitatory and 1024 inhibitory neurons. The membrane potential dynamics of neuron $j$ (excitatory or inhibitory) followed:

$$\tau_m \frac{\mathrm{d}}{\mathrm{d}t} V_j(t) = -\left(V_j(t) - V_{\text{rest}}\right) - g_j^{\text{E}}(t)\left(V_j(t) - E_{\text{E}}\right) - g_j^{\text{I}}(t)\left(V_j(t) - E_{\text{I}}\right), \tag{6}$$

where E stands for excitation and I for inhibition, $\tau_m$ = 20 ms, $V_{\text{rest}}$ = –70 mV, $E_{\text{E}}$ = 0 mV and $E_{\text{I}}$ = –80 mV.

A postsynaptic spike was emitted whenever the membrane potential $V_j(t)$ crossed a threshold $V_j^{\text{th}}(t)$, with an instantaneous reset to $V_{\text{reset}}$ = –70 mV. This threshold $V_j^{\text{th}}(t)$ was incremented by $V_{\text{spike}}^{\text{th}}$ = 100 mV every time neuron $j$ spiked and otherwise decayed following:

$$\tau_{\text{th}} \frac{\mathrm{d}}{\mathrm{d}t} V_j^{\text{th}}(t) = V_{\text{base}}^{\text{th}} - V_j^{\text{th}}(t), \tag{7}$$

with $V_{\text{base}}^{\text{th}}$ = –50 mV. The excitatory and inhibitory conductances, $g^{\text{E}}$ and $g^{\text{I}}$ evolved such that

$$g_j^{\text{E}}(t) = a g_j^{\text{AMPA}}(t) + (1 - a) g_j^{\text{NMDA}}(t) \quad \text{and}$$
$$\frac{\mathrm{d}}{\mathrm{d}t} g_j^{\text{I}}(t) = -\frac{g_j^{\text{I}}(t)}{\tau_{\text{GABA}}} + \sum_{i \in \text{Inh}} w_{ij}(t) S_i(t)$$

$$\text{with} \quad \frac{\mathrm{d}}{\mathrm{d}t} g_j^{\text{AMPA}}(t) = -\frac{g_j^{\text{AMPA}}(t)}{\tau_{\text{AMPA}}} + \sum_{i \in \text{Exc}} w_{ij}(t) S_i(t) \quad \text{and}$$

$$\frac{\mathrm{d}}{\mathrm{d}t} g_j^{\text{NMDA}}(t) = \frac{g_j^{\text{AMPA}}(t) - g_j^{\text{NMDA}}(t)}{\tau_{\text{NMDA}}}, \tag{8}$$

with $w_{ij}(t)$ the connection strength between neurons $i$ and $j$ (unitless), $a = 0.23$ (unitless), $\tau_{\text{GABA}} = 10$ ms, $\tau_{\text{AMPA}} = 5$ ms, $\tau_{\text{NMDA}} = 100$ ms, $S_i(t) = \sum \delta(t{-}t_i^*)$ the spike train of presynaptic neuron $i$, where $t_i^*$ denotes the spike times of neuron $k$, and $\delta$ the Dirac delta. Unless mentioned otherwise, all neurons received input from 5000 Poisson neurons, with 5% recurrent connectivity and constant rate $r_{\text{ext}} = 7$ Hz. The recurrent connectivity was instantiated with random sparse connectivity (10%).

## Plasticity rule parameterization

In this study, we considered three parameterizations for plasticity rules with various levels of complexity and expressivity.

**"Small polynomial" parameterization.** This polynomial search space, initially defined in [31], captured first order Hebbian spike-triggered updates:

$$\frac{\mathrm{d}}{\mathrm{d}t} w_{ij}(t) = \alpha S_i(t) + \beta S_j(t) + \gamma S_j(t) x_i(t) + \kappa S_i(t) x_j(t) \tag{9}$$

with $S_i(t) = \sum_k \delta(t - t_k^i)$ the spike train of neuron $i$, $\delta$ the Dirac delta function to denote the presence of a pre (post)-synaptic spike at time $t$. The synaptic traces $x_i$ and $x_j$ are low-pass filters of the activity of presynaptic neuron $i$ and postsynaptic neuron $j$, with time constants $\tau_{\text{pre}}$ and $\tau_{\text{post}}$, such that:

$$\frac{\mathrm{d}}{\mathrm{d}t} x_i(t) = -\frac{x_i(t)}{\tau_{\text{pre}}} + S_i(t) \quad \text{and} \quad \frac{\mathrm{d}}{\mathrm{d}t} x_j(t) = -\frac{x_j(t)}{\tau_{\text{post}}} + S_j(t), \tag{10}$$

Overall, this search space comprised 6 tunable plasticity parameters: $\theta = [\alpha, \beta, \gamma, \kappa, \tau_{\text{pre}}, \tau_{\text{post}}]$. Note that when these parameters were meta learned, the positivity constraints on the time constants $\tau_{\text{pre}}$ and $\tau_{\text{post}}$ were enforced by optimizing the natural logarithm of the time constants.

**"Big polynomial" parameterization.** Plasticity rules in this search space were parameterized such that:

$$\frac{\mathrm{d}}{\mathrm{d}t} w_{ij}(\theta, w_{ij}, x_{i,\tau_{\text{short}}}, x_{i,\tau_{\text{long}}}, x_{j,\tau_{\text{short}}}, x_{j,\tau_{\text{long}}}, \langle V_j \rangle, C_{j,E}, C_{j,I}) =$$

$$S_i(t)\theta_0 \big[1 + \theta_1 + \theta_2 w_{ij} + \theta_3 w_{ij}^2\big]\big[1 + \theta_4 \langle V_j \rangle\big]\big[1 + \theta_5 C_{j,E} + \theta_6 C_{j,E}^2\big]$$

$$\big[1 + \theta_7 C_{j,I}\big]\big[1 + \theta_8 x_{i,\tau_{\text{long}}}\big]\big[1 + \theta_9 x_{j,\tau_{\text{short}}}\big] +$$

$$S_j(t)\theta_{10}\big[1 + \theta_{11} + \theta_{12} w_{ij} + \theta_{13} w_{ij}^2\big]\big[1 + \theta_{14} \langle V_j \rangle\big]\big[1 + \theta_{15} C_{j,E} + \theta_{16} C_{j,E}^2\big]$$

$$\big[1 + \theta_{17} C_{j,I}\big]\big[1 + \theta_{18} x_{i,\tau_{short}}\big]\big[1 + \theta_{19} x_{j,\tau_{\text{long}}} + \theta_{20} x_{j,\tau_{\text{long}}}^3\big], \tag{11}$$

with $C_{j,E}(t) = \langle g_j^{\text{E}}(t)\left(E_{\text{E}} - V_j(t)\right)\rangle$ and $C_{j,I}(t) = \langle g_j^{I}(t)\left(E_I - V_j(t)\right)\rangle$ co-dependent terms representing the activity of neighboring synapses, which were low-pass filtered with fixed time constants $\tau_{C^{\text{E}}} = 10$ ms and $\tau_{C^{\text{I}}} = 100$ ms, as in previous work [20]. $\langle V_j(t) \rangle$ the low-pass filtered membrane potential, with a time-constant $\tau_{\langle V \rangle} = 100$ ms [9]. Note that, unlike for the small

polynomial search space, all timescales in this search space were not learned, and fixed to values compatible with experimental data and previous studies [9,10,12,13]. The timescales for the synaptic traces were: $\tau_{EE}^{(1)} = \tau_{IE}^{(1)} = 10$ ms and $\tau_{EE}^{(2)} = \tau_{IE}^{(2)} = 100$ ms.

Overall, this search space amounted to 21 plasticity parameters per synapse type.

**"MLP" parameterization.** In line with previous work [49], we chose a two-hidden-layer fully-connected feedforward network ("MLP"), composed of 50 sigmoidal units in the first hidden layer and 4 in the second.

In this MLP search space, the same plasticity variables as for the big polynomial were combined such that:

$$\frac{\mathrm{d}}{\mathrm{d}t} w_{ij}(t) = S_i(t)\mathrm{MLP}_{\mathrm{pre}}\left(x_{i,\tau_{\mathrm{long}}}, x_{j,\tau_{\mathrm{short}}}, w_{ij}, \langle V_j\rangle, C_{j,E}, C_{j,I}\right) +$$
$$S_j(t)\mathrm{MLP}_{\mathrm{post}}\left(x_{i,\tau_{\mathrm{short}}}, x_{j,\tau_{\mathrm{long}}}, w_{ij}, \langle V_j\rangle, C_{j,E}, C_{j,I}\right), \tag{12}$$

The input layer of the MLP was composed of 6 neurons, with the values of the relevant synaptic variables during a spike-triggered update. This layer was followed by a first fully connected hidden layer with 50 units and sigmoid non-linearity, then by another fully connected layer with 4 units and sigmoid non-linearity. The final layer was linear, fully connected. The weights of the 2 hidden layers were randomly initialized and fixed ($\sim \mathcal{U}(\frac{-1}{n_{\mathrm{inp}}}, \frac{1}{n_{\mathrm{inp}}})$, where $n_{\mathrm{inp}}$ is the number of input features at a given layer), with identical values for the on-pre and on-post MLPs. Only the weights, bias, and output learning rate of the final linear layer were trained, for a total of 4 weights + 1 bias for each MLP, as well as a common learning rate for a total of 11 plasticity parameters per plastic synapse type (see Fig 1B).

## Mean-field analysis

Within the small polynomial search space, we performed mean-field analysis on the I-E connections to link the plasticity parameters and the population firing rates at steady state $r_{\mathrm{exc}}^*, r_{\mathrm{inh}}^*$, as done in previous work [10,15,31,49]:

$$\mathrm{IE} : r_{\mathrm{exc}}^* = -\frac{\alpha r_{\mathrm{inh}}^*}{\beta + (\kappa\tau_{\mathrm{post}} + \gamma\tau_{\mathrm{pre}})r_{\mathrm{inh}}^*} \tag{13}$$

with the additional conditions $\alpha < 0$ and $\beta + (\kappa\tau_{post} + \gamma\tau_{pre})r_{inh}) > 0$ for stability.

We can perform similar derivations for the other three synapse types plastic in isolation:

$$\mathrm{EE} : r_{\mathrm{inh}}^* = -\frac{\alpha + \beta}{\kappa\tau_{\mathrm{post}} + \gamma\tau_{\mathrm{pre}}} \tag{14}$$

$$\mathrm{EI} : r_{\mathrm{inh}}^* = -\frac{\alpha r_{\mathrm{exc}}^*}{\beta + (\kappa\tau_{\mathrm{post}} + \gamma\tau_{\mathrm{pre}})r_{\mathrm{exc}}^*} \tag{15}$$

$$\mathrm{II} : r_{\mathrm{inh}}^* = -\frac{\alpha + \beta}{\kappa\tau_{\mathrm{post}} + \gamma\tau_{\mathrm{pre}}} \tag{16}$$

## Meta learning plasticity rules with evolutionary strategies

The parameters $\theta$ of the plasticity rules were optimized using an evolutionary strategy (CMA-ES [40]). It is difficult to compute usable gradients in long unrolled computational graphs, such as spiking networks, due to exploding or vanishing gradients [53]. In the case of spiking networks, simulation time-steps have to be small (0.1 ms), and total simulation times need to be long enough to give time for plasticity to carve the weights in the network at biologically

realistic timescales. Aware of the instability problems encountered in the training of learned optimizers in Machine Learning [35,53,54], we thus use evolutionary strategies instead, for their smoothing properties [53,54].

We chose CMA-ES for its robustness and low number of hyperparameters. Briefly, at meta-iteration $i + 1$, a set of $n$ plasticity rules to evaluate is generated such that:

$$\{\theta_k\}_{1 \leq k \leq n} \sim \mathcal{N}(\theta_i^*, C_i) \tag{17}$$

with $\theta_i^*$ the current best "guess" at this stage of the optimization, and $C_i$ a covariance matrix, both of which are updated at each meta-iteration based on the scores of the set of $n$ rules tested at the current meta-iteration as well as their previous values [40].

Such gradient-free optimization strategies require the simulation of many plastic spiking networks. The generation size parameter of CMA-ES $n$ was typically chosen to be twice the number of plasticity parameters, and the number of trials $N_{\text{trials}}$ over which to evaluate a single meant that every meta-iteration required the simulation of $nN_{\text{trials}}$ (values chosen between 4 and 10) separate recurrent spiking networks. Plastic networks were simulated in C++ using Auryn, a fast simulation software for spiking networks [55].

*Covariance matrix:* The covariance matrix is updated at every meta-iteration in CMA-ES. In Figs 3, 5 and 6, we only show the covariance matrix at one meta-iteration, before the loss plateaus (typically meta-iterations 10 to 20). Once the loss plateaus, we observed that most terms in the covariance became close to 1 or -1 (S2 Fig).

## Supporting information

**S1 Fig. Familiarity task loss function visualization.** Visualization of the loss function used for the familiarity task.
(PNG)

**S2 Fig. Interpretation of plasticity rules. (A)** Covariance matrices at the last meta-iterations for the optimizations shown in Fig 2 and 3. **(B)** Optimization from Fig 2C, evolution of two plasticity parameters during the optimization trajectory. Dotted line is the mean-field theoretical prediction with the non-Hebbian terms only (same analysis as in [31]). This suggests that the task is being solved mainly via the two non-Hebbian parameters, an interpretation in line with the covariance matrix visualization.
(PNG)

**S3 Fig. Another optimization with I-to-E plasticity.** More details about the optimization shown in Fig 8A.
(PNG)

**S4 Fig. Another familiarity detection optimization with I-to-I plasticity.** More details about the optimization shown in Fig 8B.
(PNG)

**S5 Fig. Familiarity detection without synaptic plasticity.** From top to bottom, raster plot of input neurons to a network identical the ones used in Fig 4 and 6, with all connections static; raster plot of excitatory neurons; raster plot of inhibitory neurons; firing rate of the excitatory population.
(PNG)

**S6 Fig. Delayed familiarity detection with flexible co-active rules.** (A) Same network and plasticity search space as in Fig 6, but the task now involves a delay between stimulus presentation and measure of the population activity. (B) Top: evolution of the plasticity parameters across meta-training. The parameters are grouped according to whether they belong to the E-to-E or I-to-E rule, and whether they are part of the weight updates triggered by a presynaptic or by a postsynaptic spike. Bottom: corresponding evolution of the loss function. Right: Network simulated with the learned rule.
(PNG)

## Acknowledgments

We would like to thank Chaitanya Chintaluri, Nicoleta Condruz and Douglas Feitosa Tomé for insightful discussions.

## Author contributions

**Conceptualization:** Basile Confavreux, Everton J. Agnes, Friedemann Zenke, Tim P. Vogels.

**Formal analysis:** Basile Confavreux.

**Methodology:** Basile Confavreux, Everton J. Agnes, Friedemann Zenke, Tim P. Vogels.

**Software:** Basile Confavreux.

**Supervision:** Everton J. Agnes, Henning Sprekeler, Tim P. Vogels.

**Visualization:** Basile Confavreux.

**Writing – original draft:** Basile Confavreux, Tim P. Vogels.

**Writing – review & editing:** Basile Confavreux, Everton J. Agnes, Friedemann Zenke, Henning Sprekeler, Tim P. Vogels.

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
