## [Decision Letter · Decision Letter 0]

25 Sep 2024

Dear Dr Confavreux,

Thank you very much for submitting your manuscript "Balancing complexity, performance and plausibility to meta learn plasticity rules in recurrent spiking networks." for consideration at PLOS Computational Biology. As with all papers reviewed by the journal, your manuscript was reviewed by members of the editorial board and by several independent reviewers. The reviewers appreciated the attention to an important topic. Based on the reviews, we are likely to accept this manuscript for publication, providing that you modify the manuscript according to the review recommendations.

Sincerely,

Jonathan David Touboul

Academic Editor

PLOS Computational Biology

Andrea E. Martin

Section Editor

PLOS Computational Biology

Reviewer's Responses to Questions

**Comments to the Authors:**

Reviewer #1: Fairly interesting work where the authors apply evolutionary algorithms to learn the plasticity rules behind networks of recurrent spiking networks. Overall, there isn't much in the spiking literature that applies gradient-free optimization methods to SNNs.

I think its more than just the enomrous parameter space that must be "trawled to elicit functions" when applying STDP rules to recurrent SNNs. Some of this isn't related to just parameters of the plasticity rules, but multi-stable solutions of the weight matrices. Ocker and Doiron (PCB, 2015) have shown that in many instances, applying these window functions to recurrent SNNs leads to instability due to the initial condition of the weights. In their 2015 paper, they show that the trajectory of the weight motifs differs depending on the initial condition for STDP rules where depression narrowly wins out.

I think the authors should flesh out with specific examples that they discuss in greater detail about the relationship between these discovered plasticity rules to previous STDP-like curves (Figure 2B for example). Do they correspond to any curves measured in vitro, or curves measured under different neuromodulators for example?

This is more of a minor point. For the MLP case, what does the pre-post protocol look like? I would also semantically just call it an ANN rather than an MLP despite. People might think you’re using actual perceptron units (non-smooth sign functions or heavisides) which likely would create additional discontinuities in your plasticity functions. I was confused about this point until I got to the methods.

I would describe synchronous regular firing patterns (line 185) as somewhat stereotypical to areas like the hippocampus. Doesn’t this just mean that your result might apply to other brain areas? It’s hard for me to judge how synchronized figure 7 is since the spike raster is so small but there seems to be an overly cortical focus to this paper when other areas don't necessarily operate on E/I balance.

Is there any reason why the authors did not incorporate terms from the loss function into the meta plasticity rule? some of the huge gains in ML have been through devising new loss functions.

I would expand on the mean-field results in the main section, it reads like too much of a throw away when it provides intuition as to why I-I connections can’t directly control E-firing rates.

I would recommend uploading code to a github repository. I did not see a code availability statement in the manuscript.

Minor Points

Line 53: Which results on degenerate solution spaces of plasticity rules?

Line 27 -Hand-hand-tuning

I think the results introduction (Lines 55 to 59) might benefit of a summary of what’s to come.

Line 285, which mean-field papers?

Reviewer #2: This paper asks the interesting question whether synaptic learning rules can be optimised so they produce a desired behaviour of recurrent spiking E/I networks. To this end, a meta-learning approach based on an evolutionary algorithm was used to optimise several plasticity rules, ranging from polynomial of different order to an MLP. The results show that this strategy can yield learning rules that stabilise network activity, and that produce networks that solve a familiarity detection task (networks responds only to previously shown patterns). While this approach is shown to work for synapse models with a limited number of parameters, the authors suggest that large models are not sufficiently constrained by the objective functions.

There are interesting results and food for thought in this well-written and presented paper. I think however that it would be nice to see if it is possible to be firmer on some of the conclusions, this would be useful both for those interested in the optimisation method, and to get a better handle on plasticity in recurrent spiking nets.

In particular, is it correct to say that failure to optimise the model was generally due to poor convergence of the method? You also mention that the objective may not constrain the optimisation sufficiently well, but should a successful optimisation not at least reduce the loss in similar ways? I feel this matters in particular for the co-active rules (where several synapse groups, E-E, I-E etc. are fit simultaneously), as I can see that optimising a single population (i.e. I-I) may not reduce loss when the parameters of the other populations are outside a functional regime.

To address these questions, it seems prudent to postulate that "simple" parametrisations should be sufficient and informative as such rules can be hand-crafted. What is the lowest order/smallest model that can satisfy the stability constraint? As far as I can see (I may have misread), the experiments for network stabilisation only show optimisation for single populations, but no joint optimisation (Fig 6 appears to show familiarity detection, not stabilisation as suggested in line 227). Can such models be fit by jointly optimising all four synapse populations?

I would expect that the degeneracy shown in Fig. 8 will also appear during optimisation of co-active rules for the stabilisation objective. It is interesting that this context that the parameter covariance is not uniform, as this indicates the presence of "important" and perhaps interpretable directions in parameter space. You seem to interpret this non-uniformity as a weak constraint on the optimisation, but can it also be interpreted in terms of robustness/flexibility? Are there tools to address this further, e.g. methods from Bayesian experimental design (note I'm not suggesting such experiments should be done for this paper)?

Additional comments:

1. Paragraph starting line 109: You say that the optimiser converges to a solution for the I->E learning rule, but the data illustrated in Fig. 4E seems to suggest otherwise - the network does not seem to respond to the inputs at all. Did you mean to say E->I rule? It seems Fig 4C describes the result you discuss in that section.

2. line 92: Should this be Fig 3D, not 3F?

3. line 129: "Since either rule (I-to-E or E-to-E)..." - should this be "Since either rule (E-to-I or E-to-E)..."? Cf. Fig. 4C

4. line 140: "...but the network dynamics were biologically implausible (Fig.6)... " - in what way, it's not clear to me from the rate plots.

**Have the authors made all data and (if applicable) computational code underlying the findings in their manuscript fully available?**

Reviewer #1: **No: **Didn't see a code availability statement in the main manuscript.

Reviewer #2: **No: **authors promise to provide code upon publication

PLOS authors have the option to publish the peer review history of their article (what does this mean?). If published, this will include your full peer review and any attached files.

Reviewer #1: No

Reviewer #2: No

Figure Files:

Data Requirements:

Reproducibility:

References:

---

## [Decision Letter · Decision Letter 1]

25 Feb 2025

Dear Dr Confavreux,

We are pleased to inform you that your manuscript 'Balancing complexity, performance and plausibility to meta learn plasticity rules in recurrent spiking networks.' has been provisionally accepted for publication in PLOS Computational Biology.

Best regards,

Jonathan David Touboul

Academic Editor

PLOS Computational Biology

Andrea E. Martin

Section Editor

PLOS Computational Biology

Reviewer's Responses to Questions

**Comments to the Authors:**

Reviewer #1: The revisions are suitable for me, and I appreciate the authors thoughtful response and recommend publication.

If they've gone back and fourth with MLP/ANN, and are back at MLP, that's fine for me since everything's defined in the manuscripts, and its a semantic point.

**Have the authors made all data and (if applicable) computational code underlying the findings in their manuscript fully available?**

Reviewer #1: Yes

PLOS authors have the option to publish the peer review history of their article (what does this mean?). If published, this will include your full peer review and any attached files.

Reviewer #1: No

---

## [Editor Report · Acceptance letter]

PCOMPBIOL-D-24-01092R1

Balancing complexity, performance and plausibility to meta learn plasticity rules in recurrent spiking networks.

Dear Dr Confavreux,

I am pleased to inform you that your manuscript has been formally accepted for publication in PLOS Computational Biology. Your manuscript is now with our production department and you will be notified of the publication date in due course.

With kind regards,

Anita Estes
